# Transforming Properties of E6/E7 Oncogenes from Beta-2 HPV80 in Primary Human Fibroblasts

**DOI:** 10.3390/ijms26115347

**Published:** 2025-06-02

**Authors:** Francisco Israel Renteria-Flores, Andrea Molina-Pineda, Ruben Piña-Cruz, Sayma Vizcarra-Ramos, Alejandra Natali Vega-Magaña, Mariel García-Chagollán, María Teresa Magaña-Torres, Rodolfo Hernández-Gutiérrez, Adriana Aguilar-Lemarroy, Luis Felipe Jave-Suárez

**Affiliations:** 1Programa de Doctorado en Ciencias en Biología Molecular y Medicina, Centro Universitario de Ciencias de la Salud (CUCS), Universidad de Guadalajara, Guadalajara 44340, Jalisco, Mexico; renteriaf33@gmail.com (F.I.R.-F.); ruben.pinacruz@gmail.com (R.P.-C.); 2División de Inmunología, Centro de Investigación Biomédica de Occidente, Instituto Mexicano del Seguro Social (IMSS), Guadalajara 44340, Jalisco, Mexico; sayvimay0510@gmail.com; 3Unidad de Biotecnología Médica y Farmacéutica, Centro de Investigación y Asistencia en Tecnología y Diseño del Estado de Jalisco A.C. (CIATEJ), Guadalajara 44270, Jalisco, Mexico; andymopi@gmail.com (A.M.-P.); rhgutierrez@ciatej.mx (R.H.-G.); 4Programa de Doctorado en Ciencias Biomédicas, Centro Universitario de Ciencias de la Salud (CUCS), Universidad de Guadalajara, Guadalajara 44340, Jalisco, Mexico; 5Laboratorio de Investigación en Cáncer e Infecciones, Centro Universitario de Ciencias de la Salud (CUCS), Universidad de Guadalajara, Guadalajara 44340, Jalisco, Mexico; alejandra.vega@academicos.udg.mx; 6Instituto de Investigación en Ciencias Biomédicas, Centro Universitario de Ciencias de la Salud (CUCS), Universidad de Guadalajara, Guadalajara 44340, Jalisco, Mexico; chagollan@academicos.udg.mx; 7División de Genética, Centro de Investigación Biomédica de Occidente, Instituto Mexicano del Seguro Social (IMSS), Guadalajara 44340, Jalisco, Mexico; maganamt@gmail.com

**Keywords:** HPV80, cervical cancer, papillomavirus, HPV, Beta-2, E6, E7, lentivirus, fibroblast model

## Abstract

Cervical cancer is the second leading cause of cancer-related death in Mexico, primarily due to persistent infection with high-risk Alpha-papillomavirus genotypes, such as HPV16 and 18. Next-generation sequencing (NGS) has revealed a high prevalence of Beta- and Gamma-HPVs, mainly Beta-2 types 38b, 80, 107, and 122, in cervical cancer samples from Mexico. Our group previously reported that HPVs 38b, 107, and 122 possess transforming properties in primary fibroblasts; however, the oncogenic potential of E6/E7-HPV80 has not yet been elucidated. For this purpose, primary human fibroblasts were transduced with E6/E7-HPV80 (FB-E6/E7-HPV80), and functional assays were conducted to evaluate changes in proliferation, metabolic activity, and cell migration. RNA-seq analysis identified differentially expressed genes (DEGs) and enriched pathways. Fibroblasts transduced with E6/E7-HPV16 (FB-E6/E7-HPV16) or empty vector (FB-pLVX) served as controls. FB-E6/E7-HPV80 extended their lifespan and exhibited increased proliferation, metabolic activity, and migration capacity. RNA-seq analysis identified 196 upregulated DEGs (such as *GPAT2*, *MST1R*, *ACAN*, *SLCO4A1*, and *CHRNA3*) and 887 downregulated DEGs (such as *KLHDC7B*, *TRIM58*, *CST1*, *FBLL1*, *INHBE*, and *TMEM132D*) shared between FB-E6/E7-HPV80 and FB-E6/E7-HPV16. Enriched pathways included p53, TNF, IL-17, apoptosis, cell cycle, etc. These findings suggest that E6/E7-HPV80 exhibits transforming capabilities that could play an important role in cervical carcinogenesis.

## 1. Introduction

Cervical cancer (CC) is the second leading cause of cancer-related deaths among women in Mexico, with a lethality rate of 47.43% [1]. Chronic infections with high-risk human papillomavirus (HPV) are the main risk factors [2]. According to the International Human Papillomavirus Reference Center, a total of 231 HPV genotypes have been identified, which include 65 genotypes from the Alpha, 55 from the Beta, 107 from the Gamma, 3 from the Mu, and 1 from the Nu groups [3]. The Alpha genus has been the most extensively studied due to its inclusion in the high-risk HPV category, which includes the causal agents of cervical cancer (CC) identified in 99% of cases [4,5]. Conversely, the biology of the other genera remains largely unknown and poorly studied, despite their greater diversity [6,7,8]. The Alpha genus has primarily been associated with mucosal tropism, while the other genera are linked to cutaneous tropism. Nevertheless, recent studies have shown a high presence of Beta and Gamma genotypes in mucosal tissues associated with CC in the Mexican population [9].

Recently, our research group identified a high prevalence of unusual HPV genotypes in both high- and low-grade cervical intraepithelial neoplasia (CIN) and CC tissue samples. These findings were obtained through next-generation sequencing (NGS), and the identified HPV genotypes fall outside the detection range of commercially available diagnostic kits. In addition to the Alpha genus, we found that the second most prevalent genus in CC samples was Beta-papillomavirus, with a frequency of 23%. The HPV types identified include 5, 20, 38b, 80, 107, and 122. Moreover, we have demonstrated that Beta-2 HPVs possess transforming properties, especially HPV122. This subtype exhibited high proliferation, migration, metabolic activity, and invasiveness properties that were significantly comparable to those of high-risk genotypes HPV16 and 18 [9,10]. Previous studies have shown that specific Beta-papillomavirus genotypes, such as HPV5, 8, 17, 20, and 38, are associated with an increased risk of squamous cell carcinoma (SCC), particularly in areas of the skin that are exposed to sunlight [7,11,12,13]. Nonetheless, there are no previous reports of the direct involvement of Beta-papillomavirus in CC or in any other tumor tissue related to mucosa. Hence, the elucidation of the carcinogenic potential of the Beta-papillomavirus genotypes with the highest prevalence in CC samples is a priority.

The Beta genus includes 53 genotypes grouped into 6 species and 3 extra genotypes of unclassified Beta-papillomaviruses [14]. Unlike those of the Alpha genus, the genomes of Beta viruses do not integrate into the host chromosome, and viral episomes may be absent from the established cancer, which is referred to as the “hit and run” effect [15,16]. Additionally, Beta HPVs lack the E5 gene. Without the E5 oncogene, the potential tumorigenic capacity of Beta HPVs mainly relies on the E6 and E7 proteins, which are generally larger than those found in Alpha-papillomaviruses. E6 and E7 from Beta HPVs retain the ability to bind to key cellular proteins, disrupting and manipulating vital signaling pathways to reprogram the cell cycle. The tumorigenic potential of Beta-papillomavirus genotypes has been linked to the binding affinity (dissociation constant or KD) of the viral E6/E7 proteins for critical cellular targets, primarily PDZ domain-containing proteins [7,17,18].

To elucidate the carcinogenic potential of beta E6 and E7 proteins, or their role as cofactors in cancer progression, it is important to study the most prevalent Beta-papillomaviruses found in CC. Particularly, it is important to examine the transforming capacity and the global impact of their E6 and E7 proteins on the transcriptome of primary normal cells. Currently, the Beta-papillomaviruses that have been most widely studied for their direct correlation or role as cofactors in SCC are HPV5, 8, 38, and 49 [13,19,20,21]. To date, the ability to immortalize primary cultures of keratinocytes or fibroblasts has been demonstrated only for Beta HPV38, 49, and 122 [10,21,22]. Nevertheless, Beta-HPVs 8, 5, 12, 15, 17, 20, 38b, and 107 have also been shown to enhance proliferation and extend the lifespan of primary cell cultures [10,19,23].

In 2020, our research group demonstrated through NGS using FAP primers that Beta-papillomavirus 80 is the third most frequent genotype (7.26%) in CC samples obtained from the Mexican population. This finding highlights the growing epidemiological significance of this genotype [10]. Therefore, the aim of this study was to elucidate the transforming properties of E6/E7-HPV80 in primary human fibroblasts.

## 2. Results

### 2.1. Integration and Expression of HPV80 E6/E7 Genes in Fibroblast Model

To confirm the integration of E6/E7 genes from HPV80 and HPV16 into the genome of the transduced human fibroblasts, PCR assays were performed using genomic DNA and E6/E7 primers described in Table 1. Specific amplicons were obtained in FB-E6/E7-HPV16 (336 bp) and FB-E6/E7-HPV80 (109 bp) (Figure 1a). Additionally, the expression of E6/E7 at the RNA level was confirmed using NGS. As shown in Figure 1b (the left panel), the number of reads per kilobase million (RPKM) in FB-E6/E7-HPV16 was 300, while in FB-E6/E7-HPV80, it was 100; however, in both cases, the same promoter was used. Figure 1b (the right panel) shows the raw reads in each model. The red arcs visualized in FB-E6/E7HPV16 depicted the two splicing variants reported for E6-HPV16.

### 2.2. Proliferation Capability of FB-E6/E7 from HPV80 and HPV16

To determine the capability of E6/E7-HPV80 in extending the lifespan of primary fibroblasts, cells were cultured for several passages. FB-pLVX and FB-E6/E7-HPV16 were also included as controls. As observed in the representative images in Figure 2, both HPV-infected models continue to grow in the cell culture even after passage 15, while the control model FB-pLVX lost its ability to proliferate before this passage.

### 2.3. Effect of E6/E7 from HPV80 on Proliferation and Metabolic Activity

The cell index was determined in real time using the xCelligence Platform and the RTCA software ver.2.3.4, which is based on impedance measurement. As depicted in Figure 3a, after 20 h, the proliferation rate between FB-E6/E7-HPV80 and FB-E6/E7-HPV16 models was significantly faster than FB-pLVX. The comparison of cell proliferation between the three models was significantly different, with the FB-E6/E7-HPV16 model being the one that achieved the highest cell proliferation index (*p* = 0.0005) when compared with FB-pLVX (negative control), followed by the FB-E6/E7-HPV80 model (*p* = 0.0003).

In addition, WST-1 assays were performed to determine metabolic activity. As depicted in Figure 3b, the same tendency was observed after 24 and 48 h, with higher metabolic activity in the FB-E6/E7-HVP16 (reaching 581.25% at 24 h and 1023.88% at 48 h), followed by FB-E6/E7-HVP80 cells (337.38% at 24 h and 491.07% at 48 h). FB-pLVX measurement was set as a control (100%).

### 2.4. Migration Capacity of Primary Fibroblasts Expressing E6/E7

The migration capacity of each cell model was evaluated using wound-healing assays (Figure 4). Photographs were taken at 0, 5, and 24. The FB-E6/E7-HPV16 cell model exhibited the highest migration rate, reaching 94.39% of wound closure after 24 h (*p* = 0.001). This was followed by the FB-E6/E7-HPV80 model, which showed 78.06% wound closure (*p* = 0.001). The FB-pLVX model showed only 62.91% of wound closure at the same time.

### 2.5. Transcriptomic Changes in FB-E6/E7-HPV80 and FB-E6/E7-HPV16 Cell Models

To determine the molecular changes driven by E6/E7 from HPV80, RNA-seq analysis was performed for all cell models, considering FB-pLVX as the negative control and FB-E6/E7-HPV16 as the positive transforming control. A bioinformatic analysis was conducted to identify the specific Differentially Expressed Genes (DEGs) up- or downregulated in FB-E6/E7-HPV80 and compared with their expression in FB-E6/E7-HPV16 (see Appendix A). As visualized in the volcano plot shown in Figure 5a, 1276 DEGs were found to be upregulated, while 1124 DEGs were identified to be downregulated in the FB-E6/E7-HPV80 model when compared with FB-pLVX. The volcano plot derived from the analysis of FB-E6/E7-HPV16 was also included for comparison (Figure 5b). To compare the transcriptional profiles of both cell models, we generated a Venn diagram illustrating the number of differentially expressed genes. Our analysis revealed 318 commonly upregulated genes and 322 consistently downregulated genes between the two models (Figure 5c). The most significantly upregulated DEGs were *DSG3*, *STON2*, *GLB1L3*, *DRAXIN*, *PLXNA4*, *GPAT2*, *MST1R*, *ACAN*, *SLCO4A1*, and *LEAP2*, while the most significantly downregulated DEGs were *ABCG1*, *KLHDC7B*, *CACNG7*, *ITIH5*, *TRIM58*, *PWWP3B*, *CST1*, *MMP10*, *FBLL1*, and *KIF5C* (Figure 6). This analysis also revealed a subset of non-coding genes, as depicted in Figure 7. Interestingly, several of these DEGs exhibited congruent regulatory patterns with those observed in the FB-E6/E7-HPV16 positive transformation model (Figure 6 and Figure 7).

### 2.6. Enriched Pathways

To gain insights into the biological and molecular pathways impacted by the DEGs observed in FB-E6/E7-HPV80, an enrichment analysis was conducted using the Molecular Signatures Databases KEGG (PathFindeR) and Hallmark (MsigDB). The KEGG database analysis clearly revealed the significant enrichment of pathways that play important biological roles in viral infections and immune host response. As illustrated in Figure 8a, the KEGG database revealed 20 main enriched pathways. Considering an adjusted *p*-value of <0.05 and its biological relevance, the most enriched pathways were the p53 signaling pathway, TNF signaling pathway, chemokine signaling pathway, rheumatoid arthritis, focal adhesion, IL-17 signaling pathway, viral protein interaction with cytokine and cytokine receptor, calcium signaling pathway, long-term potentiation, and nucleotide metabolism, among others. The same analysis was performed in the FB-E6/E7-HPV16 model, which showed that the common pathways modulated were p53 signaling and nucleotide metabolism (Figure 8b).

On the other hand, the Hallmark enrichment analysis for FB-E6/E7-HPV80 (Figure 9a) showed that among the 20 main enriched pathways, the normalized enrichment scores (NESs) were obtained for E2F targets, G2M checkpoints, MYC targets, mitotic spindle, oxidative phosphorylation, TGF-β signaling, glycolysis, unfolded protein response, apical junction, and hypoxia, among others. Based on the observation made for the FB-E6/E7-HPV16 model (Figure 9b), the following pathways were commonly regulated: E2F targets, G2M checkpoints, MYC targets, mitotic spindle, oxidative phosphorylation, TGF-β signaling, estrogen response, glycolysis, and hypoxia.

### 2.7. Predicted Protein–Protein Interaction (PPI) Network Analysis

Protein–protein interactions were predicted using the DEGs selected from RNA sequencing data, considering the following criteria: an adjusted *p*-value (*p*-adj) of <0.05 and a Log2 fold change greater than 2.0 or lower than −3.0. The analysis of the FB-E6/E7-HPV80 identified 97 nodes and 353 edges representing potential interactions among these 97 proteins. Using the Gene Ontology database for functional enrichment, we identified interactions among several protein groups based on their biological functions: 25 proteins involved in cell division with a connection strength (CS) of 0.98, 14 proteins related to mitotic nuclear division (CS: 1.21), 17 proteins associated with chromosome segregation (CS: 1.08), 12 proteins linked to microtubule cytoskeleton organization involved in mitosis (CS: 1.28), 12 proteins linked to mitotic sister chromatid segregation (CS: 1.30), 11 proteins linked to mitotic spindle (CS: 1.09), and 10 proteins related to cytokine activity (CS: 0.94), among others. Additionally, a similar analysis using the Uniprot database identified 27 proteins related to cell cycle (CS: 0.93), 22 proteins associated with cell division (CS: 0.71), and 10 proteins involved in mitosis (CS: 1.07).

The analysis predicts a first cluster of 42 proteins that strongly interact with each other (Figure 10a), most of which play key roles in proliferation, cell cycle progression, migration, and metastasis. The proteins that belong to this cluster are CDC20, RECQL4, PRC1, ASF1B, FOXM1, AURKB, BIRC5, MIK67, TPX2, CCNA1, MYBL2, and FAM83D. The second big cluster with important immune response functions was also identified, and some proteins of this cluster are IL-10, CCL3, CCL20, CCL28, and CXCL10, among others. The additional functions of proteins in this second cluster are related to growth factors, adhesion, and the extracellular matrix, among others. Finally, a small cluster containing WNT7B, SFRP2, FZD8, and NOTUM was also identified. Comparing the interactions found in FB-E6/E7-HPV16, several common protein clusters were found, and some of the shared proteins are *BIRC5*, *CDC20*, *PLK1*, *CCL28*, *ASF1B*, *MKI67*, *TPX2*, *CCNA1*, *FAM83D*, *WNT7B*, *SFRP2*, *AURKB*, *MYBL2*, *IL-10*, and *CCL28*, among others (Figure 10b).

### 2.8. Validation of DEGs Regulated in E6/E7-HPV80 and E6/E7-HPV16 Models

Based on the comparisons between the E6/E7-HPV80 and E6/E7-HPV16 models in the STRING analysis, we were interested in validating some of the genes that are important at the protein level in both models. Real-time PCRs were carried out considering FB-pLVX as a calibrator (set as 1) and FB-E6/E7-HPV16 as a transforming model control. As observed in Figure 11, all genes selected were validated in both models. *IL-10* was the most upregulated gene in the FB-E6/E7-HPV80 model, followed by *BIRC5*, *CCL28*, *ASF1B*, *FAM83D*, *MKI67*, *TPX2*, *CCNA1*, *WNT7B*, *AURKB*, and *MYBL2*. Compared to the expression levels observed in FB-E6/E7-HPV16, all genes followed the same trend. Notably, *ASF1B* and *MYBL2* expression has a higher fold change compared to that observed in the HPV80 model.

## 3. Discussion

Beta-papillomaviruses are primarily associated with skin infections and are classified as cutaneous HPVs [24,25]. However, in 2020, our research team reported a high frequency of Beta-2-HPVs in CC samples from Mexican patients [9] and subsequently demonstrated that HPVs 38b, 107, and 122 possess malignant transforming properties [10]. Nevertheless, the oncogenic properties of HPV80 have not been elucidated, despite being frequently found in Mexican CC samples (7.26%). This finding prompted us to investigate the transforming properties of the E6/E7 genes from HPV80 in primary fibroblast cultures, which have been successfully used in previous in vitro studies to demonstrate the transforming capacity of various HPVs, making them an appropriate choice for our study [9,10,26].

There are limited studies describing the prevalence of HPV80. Mackintosh et al. reported the presence of HPV80 in 6% of samples from patients with actinic keratosis (AK), intraepidermal carcinoma (IEC), or squamous cell carcinoma (SCC) [27]. Similarly, Nindl et al. reported the presence of HPV80 in 10% of actinic keratosis samples and in 7% of patients with normal skin [28]. Additionally, Sichero et al. reported the presence of a broad range of α-, β-, and γ-HPV in the male genital samples as part of the HPV infection in men (HIM) study, conducted in Mexico, Brazil, and the USA [29]. While its presence in both healthy and cancerous tissues suggests its ubiquitous nature, the specific implications of HPV80 infection in different pathologies and its potential oncogenicity warrant further research. In this study, we demonstrate that the constitutive expression of E6/E7-HPV80 genes induces a malignant phenotype marked by extended cellular lifespan (Figure 2), accelerated proliferation (Figure 3), and enhanced migratory capacity—effects (Figure 4) comparable to those observed with E6/E7-HPV16. These findings align with prior reports linking E6 and E7 proteins of Beta-papillomaviruses of HPV5, 8, 12, 15, 17, 20, 38b, and 107 to cellular transformation [10,19,23].

Previous studies have shown that E6 and E7 proteins from some Beta-papillomavirus genotypes significantly increase both the proliferation and metabolic activity of primary skin cells [10,21,30,31]. Consistent with these findings, the FB-E6/E7-HPV80 model also exhibited significantly higher proliferation and metabolic activity than the FB-pLVX control, but they were below the levels observed in the FB-E6/E7-HPV16 cell model (Figure 3).

Among the genes exhibiting the most significant upregulated expression in the FB-E6/E7-HPV80 model (as depicted in Figure 6), *DSG3* is expressed in head and neck SCC but also in breast, colorectal, and gastric cancer, implicated in the cell proliferation, metastasis, and modulation of the tumor microenvironment [32,33]. Regarding *STON2*, no association has been reported in CC; however, it has been found to be significantly increased in epithelial ovarian cancer (EOC) cell lines and tissues compared to healthy control biopsies. In addition, protein overexpression exhibits platinum resistance and has been proposed as a biomarker that predicts an unfavorable prognosis for EOC patients [34]. On the other hand, the overexpression of *STON2* has also been observed in oral squamous cell carcinoma (OSCC) tissues, and its overexpression in ovarian cancer-derived cell lines induces a higher rate of proliferation and migration [35]. In relation to *GLB1L3*, there is very limited information about its function and its expression in cancer; however, the expression levels of this gene were found to be significantly increased in invasive lung adenocarcinoma (LUAD) compared to normal tissues [36]. The high expression levels of *DRAXIN* have been observed in small-cell lung carcinoma (SCLC) [37], as well as in glioma tumor tissues and cell lines; *DRAXIN* has been proposed to function as an oncogene, as its knockdown significantly inhibits the proliferation and invasion of glioma cells [38]. Although *PLXNA4* is predominantly downregulated in several tumors [39], its inhibition in U87-MG glioma cells reduces their proliferation and tumor-forming capacity, suggesting that *PLXNA4* promotes tumor progression [40].

As observed in Figure 6, the most significant downregulated gene in the E6/E7-HPV80 model includes *ABCG1*. This gene has been found to be overexpressed in lung cancer tissues compared to adjacent normal tissues, suggesting its potential role as an oncogene in lung cancer. Additionally, studies have shown that *ABCG1* regulates markers associated with apoptosis, proliferation, migration, and invasion of lung cancer cells [41]. The observed downregulation of *ABCG1* in our model contrasts with these findings in lung cancer. This discrepancy suggests that the role of ABCG1 in cancer may vary depending on the tumor type and cellular environment.

In contrast to our observation that expression of *KLHDC7B* (Kelch domain-containing protein 7B) is significantly decreased in the presence of E6/E7 of HPVs 80 and 16 (as depicted in Figure 6), Wang et al. reported that E7 from HPV16/18 indirectly enhances *KLHDC7B* expression in HPV+ and HPV- cervical cancer cells [42]; this difference could be due to the use of different study models. On the other hand, *KLHDC7B* has been implicated in the development and progression of cervical squamous cell carcinoma (CSCC) through different mechanisms, including the alternative splicing (AS) site in the *KLHDC7B* gene, which was present in 67.5% of CSCC samples. This AS event was significantly associated with cellular differentiation and tumor size, suggesting a role in tumor progression [43]. It has also been demonstrated that the knockdown of *KLHDC7B* suppressed the tumor-promoting effects, which included increased proliferation, migration, invasion, and decreased apoptosis in CC cells [42]. *CACNG7* encodes for the calcium voltage-gated channel auxiliary subunit gamma 7 protein that regulates the activity of L-type voltage-gated calcium channels, primarily in the brain. However, there is limited information regarding its function, and the current evidence does not strongly link it to cancer development or progression. Another important downregulated gene found in the E6/E7-HPV80 model is *ITIH5*, an extracellular matrix protein that is suggested to be a tumor suppressor in various cancers. Its expression is frequently downregulated in several malignancies, including gastric, breast, bladder, and melanoma cancers, often due to epigenetic modifications such as promoter hypermethylation [44,45,46,47]. A study utilizing microarray-based expression analysis found that ITIH5 mRNA levels were significantly lower in cervical cancers compared to high-grade precursor lesions (CIN3), suggesting a role for *ITIH5* in cervical carcinogenesis [48]. *TRIM58* is an E3 ubiquitin ligase that plays a significant role in regulating various cellular processes, including tumor progression. While its involvement in CC is under investigation, studies on other malignancies provide insights into its potential functions that include being a tumor suppressor in colorectal cancer [49] or inactivating the β-catenin signaling pathway in gastric cancer, resulting in tumor growth inhibition and the enhanced apoptosis of cancer cells [50].

As observed in Figure 8, the KEGG pathway enrichment analysis performed in FB-E6/E7-HPV80 clearly revealed significant enrichment in the p53, TNF, and chemokine signaling pathways, which are widely documented to be regulated by high-risk HPVs [51,52,53,54,55]. On the other hand, as depicted in Figure 9, the Hallmark enrichment analysis provided evidence that E6/E7 from HPV80 have the potential for malignant transformation, since they impact important targets, such as E2F, Myc, G2M checkpoints, etc., which are also widely known to be affected by the presence of different HR-HPVs [56,57]. Most of the enriched pathways identified in this study play important biological roles that are closely linked to viral infections, host response, proliferation, and metabolic activity, which may provide insights into the behavior and primary mechanisms of action of the HPV80 E6/E7 genes within the host cell.

Derived from the analysis of protein–protein interactions predicted using the DEGs selected from RNA sequencing data identified in the FB-E6/E7-HPV80 or FB-E6/E7-HPV16 cell models, we found common clusters of upregulated proteins that strongly interacted with each other (Figure 10), and most of them play key roles in proliferation, cell cycle progression, migration, and metastasis. These proteins include *BIRC5*, which has been found to contribute to a reduction in apoptosis and promote cell survival, thus facilitating the progression of CC [58]. Additionally, *BIRC5* is not only involved in apoptosis regulation but also in autophagy, cell growth, and HPV infection pathways, as evidenced by RNA-seq data from the TCGA and GEO databases [59]. In recent studies, *BIRC5* expression has been shown to correlate with high-risk squamous intraepithelial lesions (HSILs), where its mRNA levels in liquid-based cytology (LBC) samples exhibited 90% sensitivity and moderate specificity (58%) [60]. Moreover, when combined with other biomarkers like *CDKN2A/p16* and *TOP2A*, diagnostic accuracy was improved, highlighting *BIRC5*′s potential as a biomarker for early CC detection. *CCL28*, a chemokine implicated in various cancers, exhibits diverse prognostic impacts depending on the cancer subtype. In pancreatic ductal adenocarcinoma (PDAC), *CCL28* is upregulated, with a negative correlation with overall survival (OS) [61]. Contrastingly, in breast cancer, *CCL28* expression has a dual effect: in luminal-like subtypes, it correlates with improved relapse-free survival (RFS) and disease-free survival (DFS), while in triple-negative breast cancer (TNBC), high *CCL28* expression is associated with poorer prognosis [62]. Furthermore, *CCL28* has been identified as a potential biomarker for predicting immunotherapy responses in lung adenocarcinoma, particularly in patients non-responsive to treatment [63]. *ASF1B* (Anti-Silencing Function 1B) is another key player in CC, with its overexpression reported in HPV-associated CC. This gene promotes the proliferation, migration, and invasion of cancer-derived cell lines by regulating *CDK9*, a key cell cycle regulator [64]. *ASF1B* silencing in CC-derived cell lines led to reduced proliferation and increased apoptosis, indicating its role in CC progression [65]. Regarding *FAM83D*, it has been shown to be overexpressed in multiple tumor types, including CC, and correlates with cell cycle progression, DNA damage response, and cell proliferation. It has been particularly implicated in breast cancer, where its expression affects tumor progression [66]. The high expression of *FAM83D* has been linked to the invasive nature of epithelial ovarian cancer, supporting its role in cancer metastasis and progression [67]. On the other hand, *MKI67*, a marker for cell proliferation, has been identified as a risk factor for HPV reinfection and CIN recurrence. The positive expression of MKI67 in cervical conization tissues correlates with disease progression, highlighting its potential as a biomarker for monitoring the progression of cervical intraepithelial neoplasia (CIN) [68]. The Targeting Protein for Xenopus Kinetochore (*TPX2*) is another critical protein involved in CC. The overexpression of *TPX2* correlates with persistent or recurrent CIN after cervical conization, and *TPX2* silencing in CC cells resulted in apoptosis, reduced cell proliferation, and invasion, demonstrating its therapeutic potential [69]. The coexpression of *TPX2* with *PD-L1* also has implications for immune evasion and cancer progression. Additionally, *CCNA1* (Cyclin A1) is involved in the regulation of the cell cycle and is subjected to promoter methylation by HPV16 E7, contributing to tumorigenesis in CC. The methylation status of *CCNA1* has been proposed as a diagnostic biomarker for grading CIN lesions, with high specificity and sensitivity in detecting high-risk lesions [70]. *WNT7B* has been identified as a critical factor in the progression of several cancers, including cervical cancer (CC). In HPV16/18-positive CC cells, WNT7B is overexpressed and packaged into extracellular vesicles (EVs), contributing to tumor angiogenesis via the activation of the β-catenin signaling pathway. Elevated levels of circulating EV-WNT7B are associated with poor clinical outcomes in CC patients, suggesting its value as a prognostic biomarker [71]. In addition, WNT7B has been shown to drive tumor proliferation and invasion through WNT/β-catenin signaling in other cancers, such as colorectal cancer [72] and oral squamous cell carcinoma [73]. Aurora kinases, a family of serine/threonine kinases, are frequently dysregulated in many types of cancer, including HPV-driven malignancies [74]. Notably, aurora kinase B (AURKB) is overexpressed in 40–60 % of HPV-associated cancers, as demonstrated in cervical and head and neck squamous cell carcinoma [75,76]. Furthermore, it has been reported that AURKB is a bona fide interacting partner of E6 from HPV16, inducing cell immortalization and proliferation [77]. *MYBL2* was identified as a key hub gene in a bioinformatic screening study aimed at discovering novel biomarkers for cervical cancer [78]. Additionally, elevated MYBL2 protein expression has been shown to serve as a predictive biomarker for progression in cervical intraepithelial neoplasia [79]. Finally, IL-10, a cytokine known for its immune-regulatory properties, has been linked to HPV infection and CC progression. Elevated IL-10 levels are associated with immune evasion and viral persistence, suggesting that it may serve as a potential biomarker for assessing disease progression in HPV-positive CC patients [80].

## 4. Materials and Methods

### 4.1. Cloning of E6/E7 Genes from HPV80

The E6/E7 Open Reading Frame (ORF) sequence of HPV80 was obtained from the Papillomavirus Episteme PaVE database (National Institutes of Health, NIH, Bethesda, MD, USA) and synthesized as a gBlock (double-stranded DNA with cohesive ends) by Integrated DNA Technologies, Inc. (IDT, Coralville, IA, USA). The lyophilized gBlock was resuspended in Tris-EDTA Buffer (pH 8.0) to obtain a final concentration of 10 ng/μL. Terminal 3’adenines in both strands were added to the insert employing the PCR DreamTaq Master Mix Kit (Cat. No. K1081, Thermo Fisher Scientific Inc., Waltham, MA, USA). The HPV80 E6/E7 ORF was initially cloned into the pGEM-T Easy vector (Cat. No. A1360, Promega, Madison, WI, USA) employing the Anza T4 DNA ligase Master Mix (Cat. No. IVGN2104, Thermo Fisher Scientific Inc., Waltham, MA, USA). Subsequently, the E6/E7 ORF of HPV80 was subcloned into the expression vector pLVX-Puro (Cat. No. 632164, Clontech Laboratories Inc., Mountain View, CA, USA) using Anza 11 EcoRI enzyme for restriction (Cat. No. IVGN0116, Thermo Fisher Scientific Inc.), Anza Antarctic Alkaline Phosphatase for vector dephosphorylation (Cat. No. IVGN2204, Thermo Fisher Scientific Inc.), and Anza T4 DNA ligase for ligation. To confirm the proper insertion of the fragments and verify the absence of mutations, the constructed vectors were sequenced using the BigDye v3.1 Terminator Cycle Sequencing Kit (Cat. No. 4337455, Thermo Fisher Scientific Inc.) and the SeqStudio Genetic Analyzer (Cat. No. A35644, Thermo Fisher Scientific Inc.). The primers used were M13 Forward and Reverse (Cat. No. N52002 and N52002, respectively, Thermo Fisher Scientific Inc.) for the pGEM_E6/E7_HPV80 construct and the Forward 5′-CCATAGAAGACACCGACTC-3′ and Reverse 5′-ATTATCTAGAGTCGCGGG-3′ for the pLVX-Puro-derived constructions. For validation, the obtained sequences were aligned with the sequences reported in the PaVE database (NIH) using the UGENE [81] software (v.48.1, Unipro). The final construct was used to generate lentiviral particles.

### 4.2. Lentiviral Particle Production, Quantification, and Viral RNA Isolation

Lentiviral vector stocks were produced by transfecting 1 μg of the gene transfer plasmid pLVX_E6/E7_HPV80, 1 μg of the pCMV-gag-pol packaging plasmid, and 1 μg of the pMD expression plasmid (envelope VSV-G) into subconfluent (7.5 × 10^5^ cells/100 mm plate) monolayer cultures of Lenti-X 293T cells (Cat. No. 632180, Takara Bio USA, Inc., San Jose, CA, USA) using the Lipofectamine 3000 transfection reagent (Cat. No. L3000001, Thermo Fisher Scientific Inc.) and incubating them at 37 °C in a 5% CO2 atmosphere for 72 h prior to the collection of lentiviral particles in the supernatant, following the protocol described by Trono Didier [82]. To quantify and validate the presence of E6/E7-HPV80 ORFs in the lentiviral particles, qPCR was performed. First, viral RNA was extracted using the Quick-RNA Miniprep Plus Kit (Cat. No. R1058, Zymo, Irvine, CA, USA) and reverse-transcribed with the Transcriptor First Strand cDNA Synthesis Kit (Cat. No. 04 379 012 00, Roche Life Sciences, Basel, Switzerland). Thereafter, qPCR was carried out using the primers described in Table 1. The quantification of lentiviral particles was achieved by generating a standard curve using the plasmid pLVX-Puro. All procedures were performed according to the manufacturers’ protocols.

### 4.3. Cell Culture

Lenti-X 293T cells and primary fibroblasts were cultivated in Dulbecco’s Modified Eagle Medium (DMEM) with 1 g/L glucose (Cat. No. 10567014, Gibco, Thermo Fisher Scientific Inc.) supplemented with 10% fetal bovine serum (FBS) (Cat. No. 35-010-CV, Corning Inc., Monterrey, Mexico) and 1% penicillin–streptomycin (100 μg/mL) (Cat. No. 15140-122, Gibco, Thermo Fisher Scientific Inc.). The cells were grown at 37 °C in a 5% CO_2_ atmosphere in a Binder incubator (Model C170UL-120V-R, Tuttlingen, Germany).

### 4.4. Fibroblast Transduction and E6/E7 Expression Evaluation

Primary fibroblasts were infected with lentiviral particles containing the E6/E7 ORFs of HPV80 and HPV16 at a multiplicity of infection (MOI) of 5. After 72 h, the culture medium was removed and replaced with a fresh medium supplemented with 0.35 μg/mL of puromycin (Cat. No. A1113803, Thermo Fisher Scientific Inc.). After 6 days, a fresh dose of puromycin was administered to select the transduced fibroblasts efficiently. Cellular models were designated as FB-E6/E7-HPV80, FB-E6/E7-HPV16, and FB-pLVX-Puro (negative control). To evaluate E6/E7 expression in the transduced fibroblasts, total RNA was extracted from the transduced models using the Quick-RNA Miniprep Plus Kit (Cat. No. R1058, Zymo, Irvine, CA, USA). cDNA was synthesized with the Transcriptor First Strand cDNA Synthesis Kit (Cat. No. 04 379 012 00, Roche Life Sciences). qPCR was performed using the LightCycler^®^ FastStart DNA Master SYBR Green I Kit (Cat. No. 03 003 230 001, Roche Life Sciences) on a LightCycler 2.0 system (Roche Life Sciences). Primers used for amplification are listed in Table 1. Actin (NC_000007.14, NCBI) was used as a housekeeping gene. To visualize the qPCR products (E6/E7 genes) for each cell model, electrophoresis was performed on a 2% agarose gel.

### 4.5. Evaluation of E6/E7 Integration in Genome of Transduced Fibroblasts

Genomic DNA (gDNA) from all cell models was extracted using the DNeasy Blood and Tissue Kit (Cat. No. 69504, Qiagen, Hilden, Germany). A total of 100 ng of gDNA was used as a template, and qPCR was performed on the LightCycler 2.0 system using the LightCycler^®^ FastStart DNA Master SYBR Green I reagents described above and primers listed in Table 1. Amplicons were visualized in 2% agarose gels.

### 4.6. Immortalization Assay

A total of 300,000 cells from each model were seeded on T75 flasks and photographed every 4–5 days (upon reaching 80–90% confluence) using a Primo Vert inverted microscope (Model No. 415510-1101-000, Carl Zeiss Microscopy GmbH, Jena, Germany) with a 10X objective. Subsequently, cell passaging was performed by treating the cells with 1 mL of 0.25% Trypsin–EDTA (Cat. No. 25200056, Thermo Fisher Scientific Inc.) and neutralizing it with the supplemented DMEM. This procedure was repeated until passage 20.

### 4.7. WST-1 Metabolic Activity Assay

Five thousand cells from each model were seeded in quintuplicate into 96-well plates, with each well containing 100 μL of the supplemented DMEM. A 10 μL volume of WST-1 reagent (Water Soluble Tetrazolium Salt-1, Cat. No. 5015944001, Roche Life Sciences) was added after 24 h and 48 h, respectively. Absorbance at 450 nm was measured after 5 h using a Synergy HT multimode microplate reader (Cat. No. 7091000, BioTek Instruments, Inc., Winooski, VT, USA). The absorbance values measured for FB-pLVX cells were set as the 100% reference to determine the percentage of metabolic activity in each cell model.

### 4.8. Cell Proliferation Assay

To perform the cell proliferation assay, 3000 cells from each model were seeded in a 96-well E-Plate (Cat. No. 05232368001, Agilent Technologies, Inc., Santa Clara, CA, USA). Cell proliferation was measured every hour for 48 h using the Real-Time Cell Analyzer (RTCA) xCelligence System (Cat. No. 05228972001, Agilent Technologies, Inc.), and the results were depicted as the cell index.

### 4.9. Cell Migration Assay

In order to evaluate the migration capacity of the established models, 300,000 cells from each cell line were seeded in duplicate into 6-well plates (Cat. No. 140675, Thermo Fisher Scientific Inc.) and incubated for 24 h with complete DMEM (3 mL/well) until they reached 70–80% confluence. After that period, the old medium was removed, and cells were treated for 2 h with 3 mL of complete DMEM supplemented with 1.25 μg/mL of mitomycin C (Cat. No. 10107409001, Roche Life Sciences) to suppress cell proliferation. Afterward, three scratches were manually made on each well. Cells were washed with PBS (phosphate-buffered saline) (Cat. No. 10010023, Thermo Fisher Scientific Inc.) to remove residual mitomycin C and debris, and fresh complete medium was added to each well. Photographs were obtained with a 10× objective at 0, 5, 24, and 30 h using a Primo Vert inverted microscope (Carl Zeiss Microscopy GmbH), with a ZEISS AxioCam camera (Model ERc5s, Carl Zeiss Microscopy GmbH), using the ZEN 2012 software (Blue Edition, Carl Zeiss Microscopy GmbH). The percentage of wound closure was determined by measuring the gaps with the use of ImageJ ver. 1.54 [83] (Image Processing and Analysis in Java; Rasband, W.S., ImageJ, U.S. National Institutes of Health, Bethesda, MD, USA, https://imagej.nih.gov/ij/ last accessed on 30 May 2025) and the Wound Healing Size Tool an ImageJ/Fiji^®^ plugin (https://github.com/AlejandraArnedo/Wound-healing-size-tool/wiki, last accessed on 30 May 2025), that allows the quantification of wound area, wound coverage of total area, average wound width, and width standard deviation in images obtained from a wound healing assay [84].

### 4.10. RNA-Seq

To assess the quality of the extracted RNA of all our models, the Agilent RNA 6000 Nano Kit (Cat. No. 6067-1511) and the Bioanalyzer 2100 System (Agilent Technologies) were used. Samples with an RIN score higher than 8 were considered for sequencing. A total of 3.0 µg of RNA of each sample was sent to and analyzed by Novogene Bioinformatics Technology Co., Ltd., in Beijing, China. Library construction and NovaSeq 6000 Illumina sequencing were performed by Novogene from each cell model, using 4 independently extracted RNAs for HPV80 and 6 for pLVX-puro cell models. The raw data obtained from sequencing were uploaded to the GeoData repository with the accession number GSE279652, which will be released on June 30, 2025. The raw sequencing data files were subjected to quality control using the FastQC Read Quality Reports Tool (Galaxy, Ver. 0.74 + galaxy0) available on the Galaxy platform (https://www.usegalaxy.org, accessed on 4 September 2024). All raw data files with a minimum Phred score of 36 were subsequently processed and analyzed using R software (ver. 4.4.1) and RStudio (2024.09.0 + 375) with the Rsubread package (v2.14.2) [85]. The reference genome index was built using the Genome Reference Consortium Human Build 38 patch release 14 (GRCh38.p14) version 46 (https://www.gencodegenes.org/human/release_46.html released on May 2024; last accessed on 6 March 2025). Reads were mapped and aligned to the human reference genome using the Align function of Rsubread. The resulting BAM files were processed to generate gene count files through the FeatureCounts function (v2.0.3), and a read count matrix was constructed. Differentially expressed genes (DEGs) were identified using the DESeq2 package (ver. 1.42.0), comparing the HPV80 or HPV16 groups with the pLVX group (Appendix A). Finally, enrichment analysis was performed using the PathFindeR package (ver. 2.3.1) [86], employing gene sets from the Kyoto Encyclopedia of Genes and Genomes (KEGG). Additionally, Gene Set Enrichment Analysis (GSEA) was performed using the Hallmark pathways from the Molecular Signatures Database (MSigDB).

### 4.11. Protein–Protein Interaction (PPI) Network Analysis

Differentially expressed genes (DEGs) were selected from RNA sequencing (RNA-seq) data, considering the following criteria: an adjusted *p*-value (*p*-adj) of <0.05 and a Log2 fold change greater than 2.0 or lower than −3.0. The selected DEGs were then uploaded to the STRING database (https://string-db.org/; accessed on 14 February 2025) to construct a protein–protein interaction (PPI) network. The analysis was performed using the “Multiple Protein Interaction” option, selecting Homo sapiens as the reference organism. To ensure a high-confidence network, only interactions with a minimum confidence score of 0.7 were included. Proteins that were not connected were excluded to improve visualization and focus the analysis on functionally relevant interactions.

### 4.12. qPCR Validation of PPI Network Analysis

To validate the expression of the genes identified in the PPI network analysis, real-time qPCR was performed using the reference genes RPLP0 and RPS18 and the target primers listed in Table 2. qPCR was carried out on the LightCycler 2.0 system using the LightCycler^®^ FastStart DNA Master SYBR Green I reagents described above. All primers used were designed using the Primer-BLAST tool provided by NCBI-NIH (https://www.ncbi.nlm.nih.gov/tools/primer-blast/index.cgi accessed on 30 May 2025) using as a search database the Homo sapiens Refseq mRNA. Primer secondary structures were corroborated by the IDT OligoAnalyzer Tool (https://www.idtdna.com/pages/tools/oligoanalyzer accessed on 30 May 2025). Primer specificity and annealing conditions were determined by conventional PCR.

### 4.13. Statistical Analysis

All experiments were performed independently in duplicate with at least two repetitions. Data analysis was performed using the R program (V. 4.4.1) and GraphPad Prism (v. 8.0.1). The results of the cell tests (proliferation and metabolic activity) were expressed as mean ± standard deviation. The normality and homoscedasticity of the samples were obtained using the Shapiro–Wilk and Bartlett tests, respectively. The significant differences between samples were obtained with the analysis of variance (ANOVA) and the Tukey tests (parametric data) or with the Kruskal–Wallis and the Bonferroni tests (non-parametric data). In the proliferation assay, significant differences between cellular models were obtained using the Mann–Whitney U statistical test. For the analysis of differentially expressed genes, we considered genes as significant if they met the following criteria when compared to the FB-pLVX model: an adjusted *p*-value (adj-*p*) of less than 0.05 and a Log2 fold change (Log2FC) greater than 2 (for upregulated genes) or less than −2 (for downregulated genes).

## 5. Conclusions

Our findings provide evidence that HPV80, a Beta-2 human papillomavirus, possesses significant oncogenic properties comparable to those of HPV16. The constitutive expression of HPV80 E6/E7 genes induced a malignant phenotype in primary fibroblasts, characterized by extended cellular lifespan, increased proliferation, and enhanced migratory capacity. Although HPV80 has traditionally been classified as a cutaneous virus, its detection in cervical cancer samples from Mexican patients and its demonstrated ability to drive malignant transformation in vitro suggest that it plays a potential role in cervical cancer development. It may contribute to cervical cancer development through mechanisms like those observed in HR-HPV genotypes, involving the dysregulation of genes and pathways related to immortalization, apoptosis, cell cycle control, immune evasion, DNA repair, metabolism, migration, and epithelial–mesenchymal transition. While HPV80 shares some functional similarities with high-risk Alpha-HPVs, it also exhibits unique features not previously described in other Beta-papillomaviruses. Our findings highlight HPV80 as a significant risk factor of carcinogenesis and emphasize the need to include it in cervical cancer screening and prevention strategies. Additional research is required to fully elucidate the molecular mechanisms of HPV80 in cervical carcinogenesis and to assess its clinical relevance across diverse populations.

## Figures and Tables

**Figure 1 ijms-26-05347-f001:**
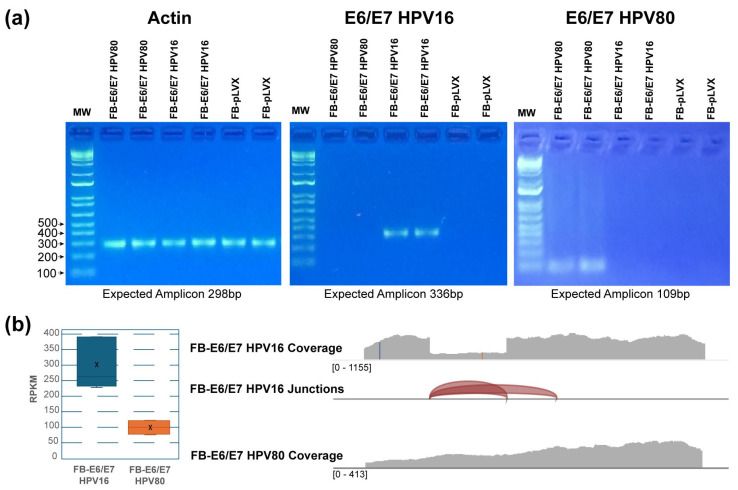
The integration and expression of E6/E7 from HPV16 and HPV80. (**a**) The visualization of E6/E7 amplicons from HPV16 and HPV80 on agarose gels. The amplification of actin was included as a DNA amplification control. The expected amplicon sizes are described below each gel. 1Kb plus DNA ladder was used as a molecular weight marker. (**b**) The chart on the left displays the number of reads (RPKM), and X indicates the mean. The image on the right displays the total coverage of the reads. Splicing junctions detected in the sequence are represented as red arcs.

**Figure 2 ijms-26-05347-f002:**
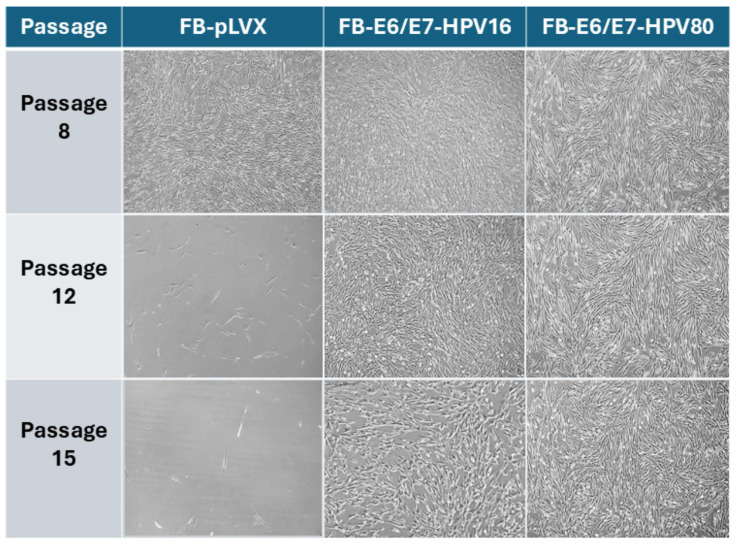
The microscopic imaging of transduced primary human fibroblast models over time. The representative microscopic images of FB-pLVX (primary fibroblasts transduced with an empty vector considered a control), FB-E6/E7-HPV16 (a positive transforming control), and FB-E6/E7-HPV80. The photographs represent passages 8, 12, and 15 using a 10× objective.

**Figure 3 ijms-26-05347-f003:**
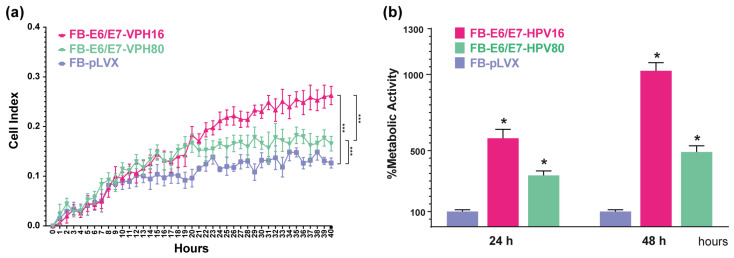
The proliferation and metabolic activity induced by the E6/E7 from HPV16 and HPV80 in primary fibroblasts. (**a**) The graph represents cell proliferation curves represented by cell index values using the xCelligence RTCA platform measured in real time for 40 h. Asterisks indicate statistical significance, *** *p* < 0.0005. (**b**) The graph depicts the percentage of metabolic activity assessed using the WST-1 assay at 24 and 48 h in all cell models. The data obtained for FB-pLVX were considered to be 100% metabolic activity. Asterisks indicate statistical significance, * *p* < 0.05.

**Figure 4 ijms-26-05347-f004:**
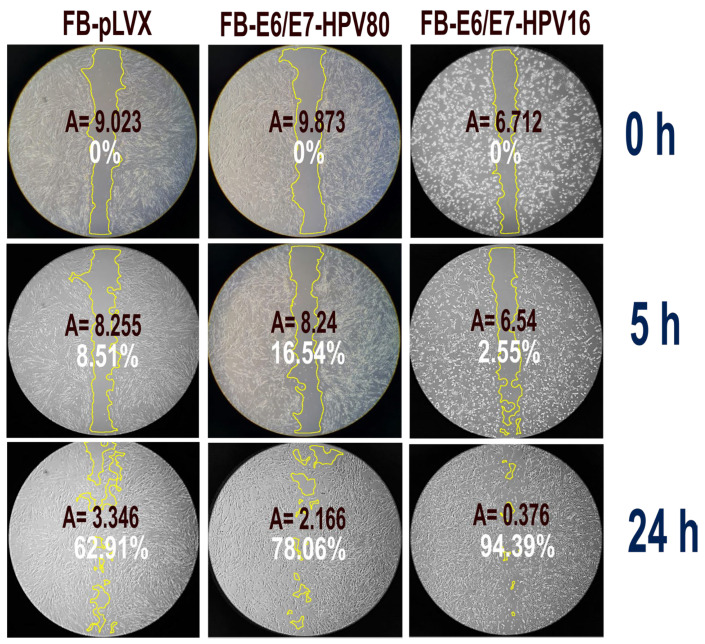
Wound healing assay. The figure depicts the percentage of wound closure in the different cell models at 0, 5, and 24 h (white label). The wound area that was measured at each time point is circled in yellow. A: size of the measured area (square inches).

**Figure 5 ijms-26-05347-f005:**
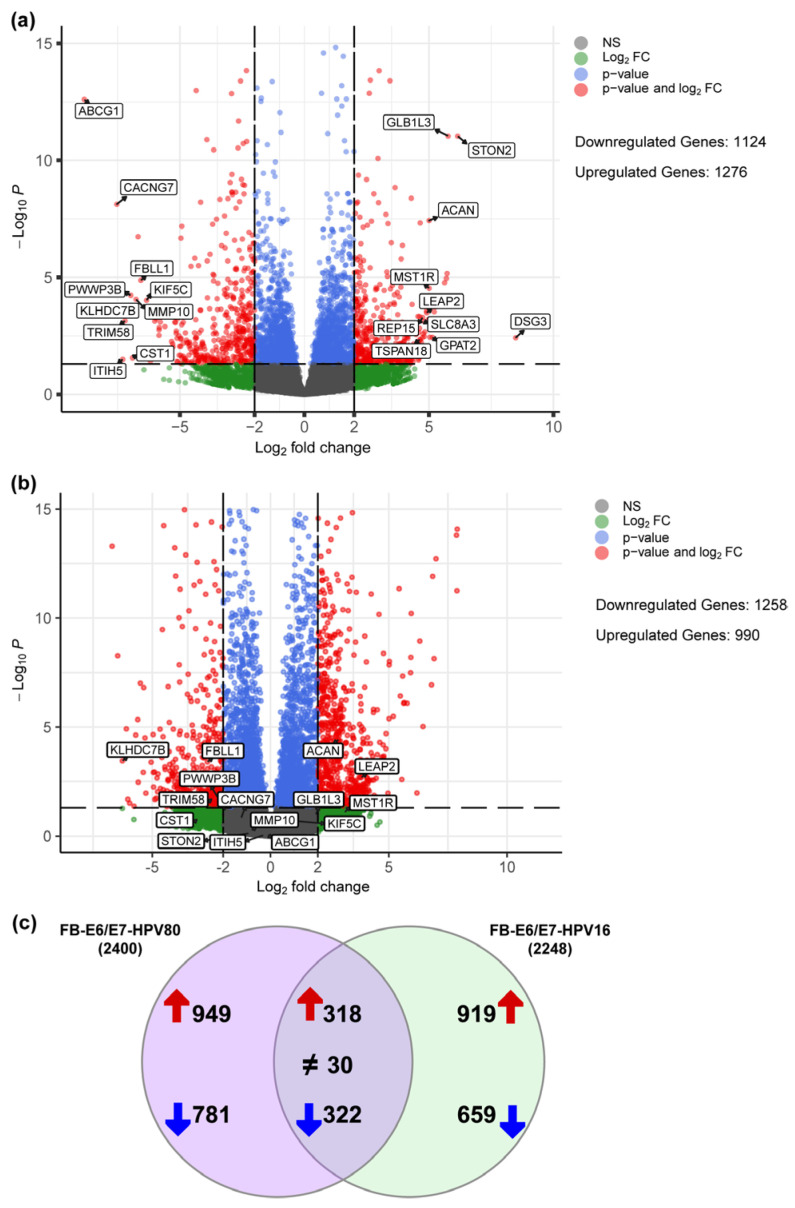
Differentially expressed genes regulated by E6/E7-HPV80 and E6/E7-HPV16. Volcano plots showing genes that are differentially downregulated (<−2 Log2FC) or upregulated (>2 Log2FC) in the fibroblast model expressing (**a**) E6/E7 from HPV80 compared with the FB-pLVX model and (**b**) E6/E7 from HPV16 compared with FB-pLVX. The *Y*-axis shows the Log10 *p*-value, and the *X*-axis shows the Log2 fold change. The highest or lowest fold change genes are labeled. (**c**) Venn diagram depicting the distribution of DEGs between FB-E6/E7-HPV80 and FB-E6/E7-HPV16. Both datasets were processed with identical threshold criteria (Log2FC ≥ 2 or ≤–2). The diagram illustrates the transcriptional overlap as well as the distinct DEGs induced by E6/E7 from each model. Upregulated genes are depicted with red arrows, while downregulated genes are represented by blue arrows. Common DEGs displaying contrasting expression patterns are annotated with a ≠ symbol.

**Figure 6 ijms-26-05347-f006:**
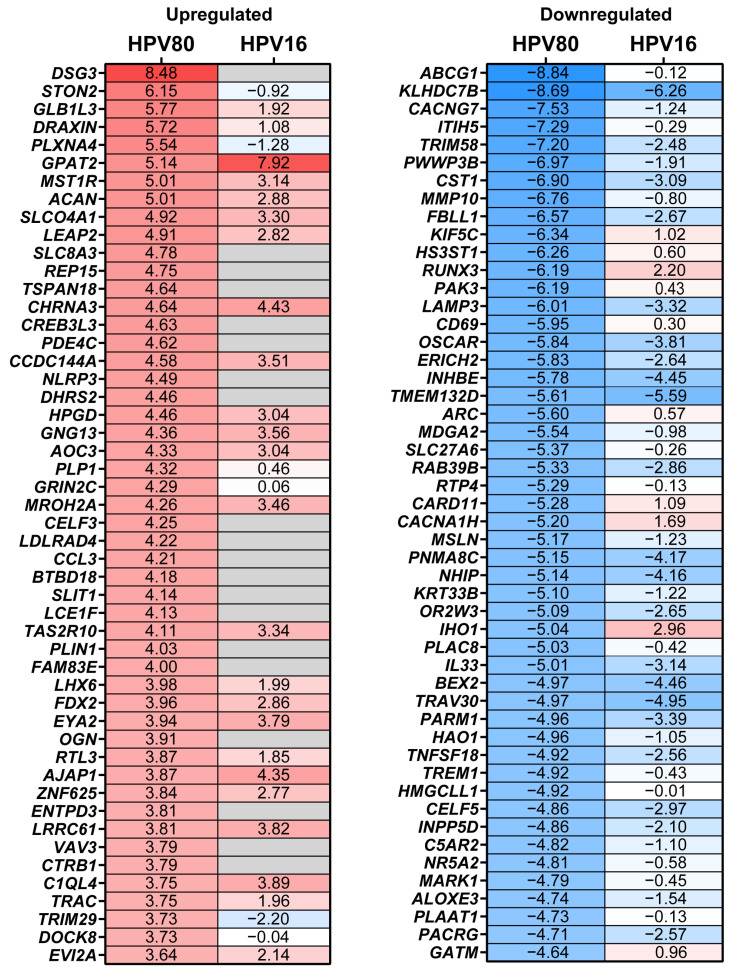
DEGs modulated by the expression of E6/E7 from HPV80. The heatmap of the top 50 up- and downregulated DEGs in the FB-E6/E7-HPV80 cell model compared to those in the FB-pLVX cell model (See Appendix A). The fold change obtained in the FB-E6/E7-HPV16 vs. the FB-pLVX model is also shown. The higher values of fold change are shown in red, and the lower values are shown in blue; gray boxes indicate genes whose differential expression was not significant (*p* > 0.05).

**Figure 7 ijms-26-05347-f007:**
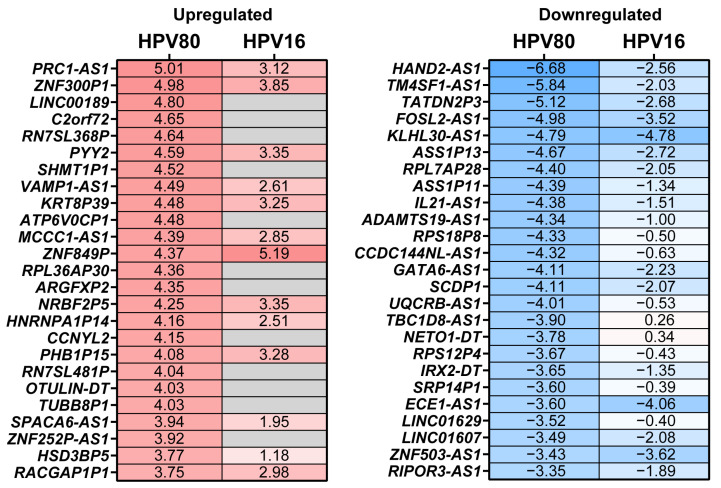
Non-coding DEGs modulated by the expression of E6/E7 from HPV80. The heatmap of the top 25 up- and downregulated non-coding DEGs in the FB-E6/E7-HPV80 cell model compared to those in the FB-pLVX cell model (See Appendix A). The fold change obtained in the FB-E6/E7-HPV16 vs. the FB-pLVX model is also shown. The higher values of fold change are shown in red, and the lower values are shown in blue; gray boxes indicate genes whose differential expression was not significant (*p* > 0.05).

**Figure 8 ijms-26-05347-f008:**
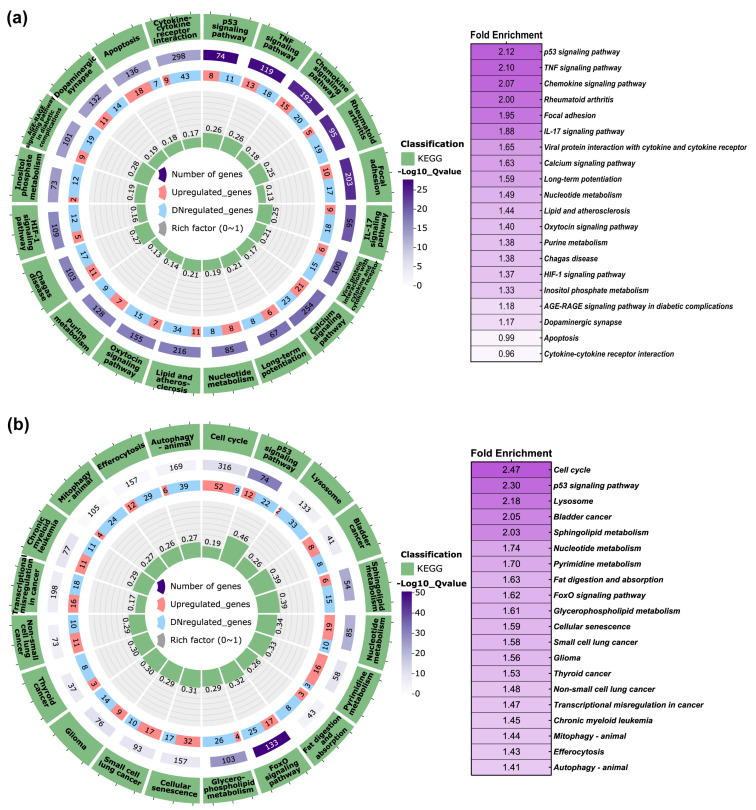
KEGG pathway enrichment analysis in FB-E6/E7-HPV80 and FB-E6/E7-HPV16. Polar plots generated using PathFindeR showing the predicted KEGG (Kyoto Encyclopedia of Genes and Genomes) pathways regulated by the presence of E6/E7 genes from (**a**) HPV80 and (**b**) HPV16. Green labels indicate enriched pathways, while purple labels denote the number of genes involved in each pathway (color intensity represents the adjusted *p*-value for controlling the false discovery rate (FDR). Red and light blue labels indicate upregulated and downregulated genes within each enriched pathway, respectively. Central green columns represent the proportion of enriched genes relative to the total gene counts in each pathway. Fold enrichment values are displayed in the table to the right of each plot.

**Figure 9 ijms-26-05347-f009:**
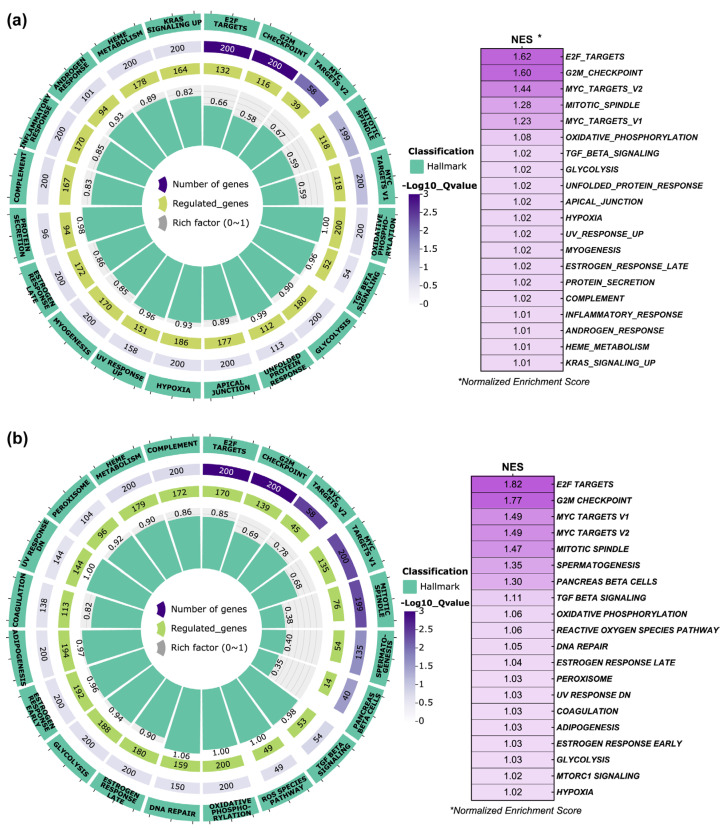
Hallmark pathway enrichment analysis in FB-E6/E7-HPV80 and FB-E6/E7-HPV16. Polar plots generated using GSEA (Gene Set Enrichment Analysis) with predefined Molecular Signatures Database (MSigDB) showing the predicted Hallmark pathways regulated by the presence of E6/E7 genes from (**a**) HPV80 and (**b**) HPV16. Green labels indicate enriched pathways, while purple labels denote the number of genes involved in each pathway (color intensity represents the adjusted Q-value controlling the false discovery rate, FDR). Central green columns represent the proportion of enriched genes relative to the total gene counts in each pathway. Normalized Enrichment Score (NES) values are displayed in the table to the right of each plot.

**Figure 10 ijms-26-05347-f010:**
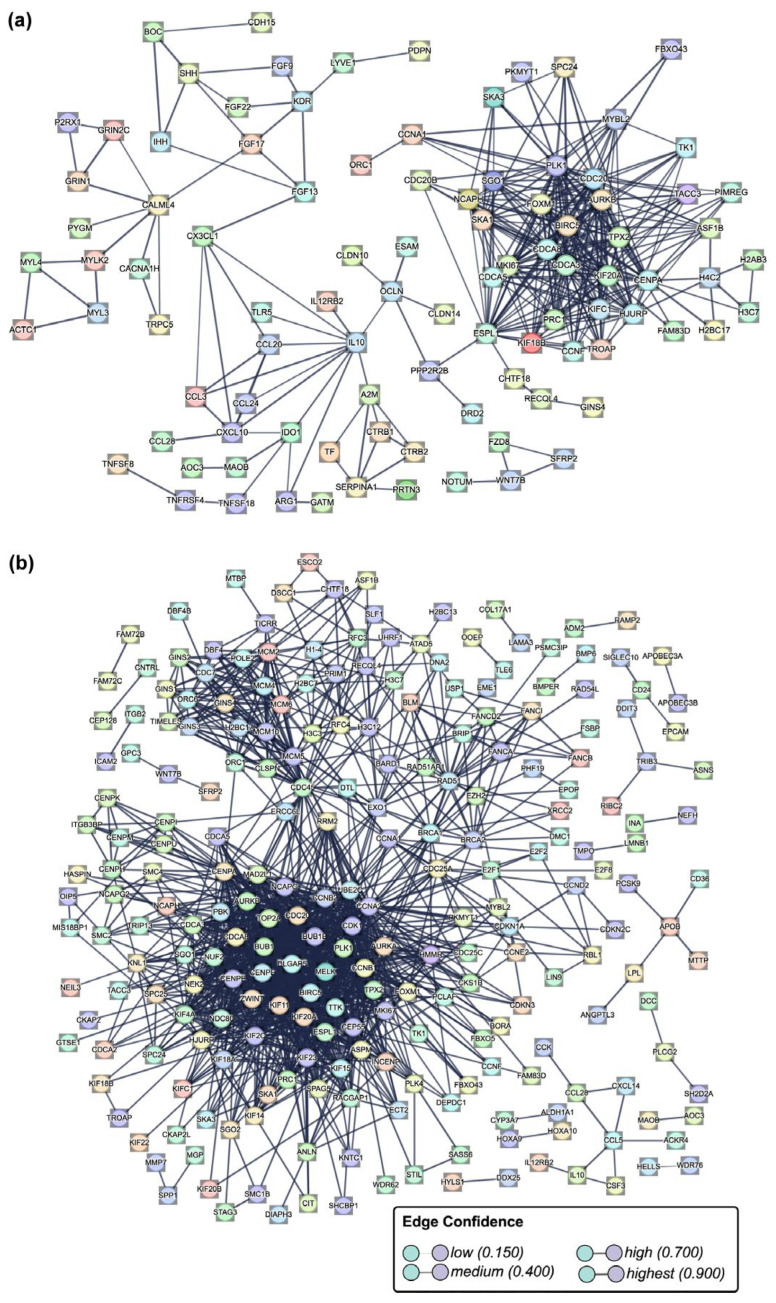
Protein–protein interaction network analysis. The network represents the interactions among differentially expressed genes (DEGs) identified in the (**a**) FB-E6/E7-HPV80 and (**b**) FB-E6/E7-HPV16 cell models using the software STRING-db (v.12.0). The nodes correspond to proteins, and the edge thickness corresponds to the confidence score of the interaction, with darker edges indicating stronger predicted associations. Highly connected clusters suggest biological processes significantly enriched in the dataset.

**Figure 11 ijms-26-05347-f011:**
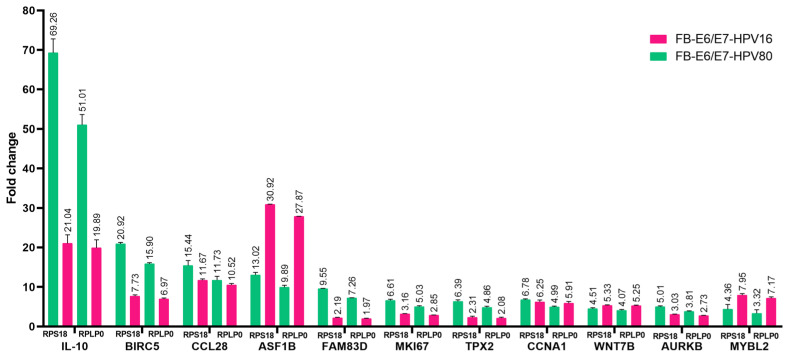
The validation of diverse DEGs using qPCR. Real-time PCR assays were performed in the FB-E6/E7-HPV80 and -HPV16 cell models. The FB-pLVX value of each gene was considered the calibrator (set as 1), and *RPS18* and *RPLP0* were used as the reference genes. Fold change was assessed using the 2^−ΔΔCp^ analysis.

**Table 1 ijms-26-05347-t001:** Primer sequences used to evaluate E6/E7 expression with qPCR.

Cellular Model	Forward Primer (5′-3′)	Reverse Primer (5′-3′)	Amplicon Size
HPV80	TTGAGGTAGTTGAGCGAAAAG	TGCTGCTGGTTGTAACAAAT	109 bp
HPV16	TAGAGAAACCCAGCTGTAATCA	AGGATCAGCCATGGTAGATTAT	336 bp
pLVX-Puro	GCCCGCCTTCCTGGAGACCTC	TGATTGTTCCAGACGCGGTCA	150 bp
Actin	TCCGCAAAGACCTGTACG	AAGAAAGGGTGTAACGCAACTA	410 bp

**Table 2 ijms-26-05347-t002:** Primer sequences used to evaluate expression of DEGs.

Gene	Forward Primer (5′-3′)	Reverse Primer (3′-5′)	Amplicon Size
IL-10	GCTACGGCGCTGTCATCGATT	CAGAGCCCCAGATCCGATTTTG	244 bp
BIRC5	GCTGGGAGCCAGATGACGACC	CGATGGCACGGCGCACTTT	279 bp
CCL28	AGCTGTTGCACGGAGGTTT	TTCTTGGCAGCTTGCACTTTC	191 bp
ASF1B	ACAGGAGTTCATCCGAGTGG	GCCTGTCCATGTTGTTGTCC	170 bp
FAM83D	GACAGTTCGGACTATCACAGGA	CAACCACTTGGCCAGACAGA	185 bp
MKI67	GAAAGGGAAAGGAGAAGCAGGA	TCTTGACACACACATTGTCCTCA	173 bp
TPX2	GGCCAGACTACAGGAAG	ACAGCTGAGTTTAGCAGTGGA	182 bp
CCNA1	TGACAGTACCAACCACCAACC	TGCAGCTATCAGTGAAGGAAGA	152 bp
WNT7B	GCGTTACGGCATCGACTTCT	TGTCTCCATGGGCTTCTGATAG	331 bp
MYBL2	CCGAAGCCACTTCACGACA	ACCCTCAACACCTCAGGACA	169 bp
AURKB	TGTGGTGCATTGGAGTGCTT	AGGGGTTATGCCTGAGCAGT	171 bp

## Data Availability

The datasets generated and analyzed during the study are deposited in the Gene Expression Omnibus—NCBI (accession number: GSE279652).

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
