# Peer review of "Transforming Properties of E6/E7 Oncogenes from Beta-2 HPV80 in Primary Human Fibroblasts"

_ijms, 2025, doi:10.3390/ijms26115347_

Round 1

Reviewer 1 Report

Comments and Suggestions for Authors

Renteria-Flores et al. investigate oncogenic potential of HPV80 E6 and E7 proteins (E6/E7). They used a lentiviral vector to transfect the E6 and E7 genes into primary human fibroblasts, and performed tests to elucidate altered cellular life span (immortalization), proliferation, metabolism and migration. HPV80 E6E7 altered all these parameters in comparison untransfected controls, and in similar direction as the HPV16 E6/E7 analogs, which supports the idea that these two proteins have transforming and by consequence can at least support oncogenesis. These are interesting results but not groundbreaking, as similar findings have been made for E6 and E7 proteins from other HPVs.  

Next the authors scratched the surface of the cellular mechanisms of these alterations, by performing RNAseq transcriptomics and enrichment analysis. These analyses highlighted cellular pathways that seem to be in line with what we would have expected (nucleotide and glucose metabolism, and various immune signaling cascades. Interestingly different genes were deregulated in comparison to HPV16 E6/E7. It would make sense to provide Venn diagrams to illustrate commonly deregulated features (genes, gene modules and pathways) for HPV80 and HPV16 as well as inspect the acutal enrichment values from a HPV16-centric perspective (as done in the HPV80-centric Figures 6, 7, 8, 9 and 10).  

Then the authors jumped into protein-protien interaction networks and attempted to validate with PCR. This is where things become unclear. How come the top 3 genes found as most upregulated in Figure 11 do not appear in say Figure 6? This is worrysome, and I believe the authors should revisit their data, and make these results less doubt-inducive.  

Author Response

RESPONSE TO REVIEWERS' COMMENTS

Thank you for all the comments. We appreciate the reviewer’s insightful suggestions; they allowed us to make a better version of the manuscript. In addition, language editing was performed using MDPI’s Author Services. All changes suggested by the Editor and reviewers were made, as described below:

Reviewer 1

Renteria-Flores et al. investigate oncogenic potential of HPV80 E6 and E7 proteins (E6/E7). They used a lentiviral vector to transfect the E6 and E7 genes into primary human fibroblasts, and performed tests to elucidate altered cellular life span (immortalization), proliferation, metabolism and migration. HPV80 E6E7 altered all these parameters in comparison untransfected controls, and in similar direction as the HPV16 E6/E7 analogs, which supports the idea that these two proteins have transforming and by consequence can at least support oncogenesis.

These are interesting results but not groundbreaking, as similar findings have been made for E6 and E7 proteins from other HPVs.  

Response:

While there are numerous studies related to the functions of E6 and E7 in alpha-papillomaviruses, the study of these genes in beta-papillomaviruses is very limited. Notably, no studies to date have investigated the biological or oncogenic functions of E6 and E7 in HPV80, a member of the beta-papillomavirus genus. We believe this is the relevance of the present study.

Next the authors scratched the surface of the cellular mechanisms of these alterations, by performing RNAseq transcriptomics and enrichment analysis. These analyses highlighted cellular pathways that seem to be in line with what we would have expected (nucleotide and glucose metabolism, and various immune signaling cascades. Interestingly different genes were deregulated in comparison to HPV16 E6/E7. It would make sense to provide Venn diagrams to illustrate commonly deregulated features (genes, gene modules and pathways) for HPV80 and HPV16 as well as inspect the acutal enrichment values from a HPV16-centric perspective (as done in the HPV80-centric Figures 6, 7, 8, 9 and 10).  

Response:

In the previously submitted version, HPV16 data was excluded in some figures, due to the extensive existing literature on this well-characterized alpha-papillomavirus. However, following your recommendation, we have incorporated comparative Venn diagrams in the revised manuscript to highlight the shared deregulated genes between HPV80 and HPV16 (Figure 5). Furthermore, the enrichment and STRING analyses have been expanded to incorporate direct comparisons with HPV16 (Figures 8, 9 and 10).

Then the authors jumped into protein-protien interaction networks and attempted to validate with PCR. This is where things become unclear. How come the top 3 genes found as most upregulated in Figure 11 do not appear in say Figure 6? This is worrysome, and I believe the authors should revisit their data, and make these results less doubt-inducive.  

Response:

We understand your concern and to clarify your doubts, we have the following explanation: RNA-seq analysis, as visualized in the volcano plot of the HPV80 model, identified 1,276 significantly overexpressed genes. Given the large number of candidates, we only displayed the top 50 most highly overexpressed genes in Figure 6. To systematically prioritize biologically relevant targets from the full set of overexpressed genes, we performed STRING network analysis to identify those with the most extensive protein-protein reported interactions. It may be possible that some of these genes do not appear in the STRING network analysis due to a lack of strong predicted associations, although they were identified among the top 50 DEGs.

Based on these interaction networks, in this new version of the manuscript in which the STRING analysis of the E6/E7-HPV16 model was added in Figure 10, we identified the most prominently represented genes common to both models and selected those within key clusters for experimental validation by qPCR. We identified some genes that, despite being differentially expressed, were not depicted in the STRING analysis in both models, so, in the new Figure 11 they were eliminated (CCL20, CES1, and EVI2A). Additionally, we included the expression of WNT7B, which was present in both STRING models. For this reason, Figure 11 was also modified.

To have a complete overview of all DEGs and to avoid any doubts, in the new submission, we have included supplementary data containing the transcriptomic dataset, including fold change values and statistical significance for all genes analyzed in both models. Tables in Excel format, Table S1: Differential gene expression analysis of HPV80 (DESeq2 results); Table S2: Differential gene expression analysis of HPV16 (DESeq2 results).

Reviewer 2 Report

Comments and Suggestions for Authors

The submission entitled "Transforming properties of the E6/E7 oncogenes from Beta-2 HPV80 in primary human fibroblasts" describes an interesting research work, focusing on the transforming properties from HPV80 in primary fibroblasts. In general, the text is well-written, and the results are sound to the applied methodology and scope of the work.

Major revisions

1.- In Figure 4, the authors mention that Mitomycin was used to suppress cell proliferation; this does not look clear in the figure because in FB-E6/E7- HPV16 model, cell proliferation increase a 30% during 24 h (0h vs 24h). Is necessary to repeat the Wound healing assay with a higher concentration of Mitomycin to stop cell proliferation in all groups.

The quality of these pictures should be improved. It will be interesting if the authors can show the pictures of scratch wound healing assay in higher magnification.

2.- The manuscript is overloaded with references and many of them are unnecessary. It is recommended that the references be removed and updated.

3.- Discussion is too long, some reductions are recommended.

Minor revisions

1.- In introduction, is necessary to explain if E2 protein regulates the expression of the HPV80 E6 and E7 oncogenes during cervical carcinogenesis.

2.- In Figure 6 and Figure 7.

If FB-E6/E7-HPV16 model is considered as positive control, ¿Why FB-E6/E7-HPV 80 model presented the most up- or downregulated gene expression in comparison with FB-E6/E7-HPV16?

3.- In line 264 (Figure legend), please incorporate “Predicted”.

4.- In 4.7 WST-1 Metabolic activity assay (Materials and Methods).

Commonly, Tetrazolium Salt is used in proliferation assays. ¿Why you consider that Tetrazolium Salt is a Metabolic activity assay?

5.- It is necessary to describe in the 4.12. section if the primers designed for qPCR were analyzed to identify potential secondary structures (self-dimer or hetero-dimer).

Author Response

RESPONSE TO REVIEWERS' COMMENTS

Thank you for all the comments. We appreciate the reviewer’s insightful suggestions; they allowed us to make a better version of the manuscript. In addition, language editing was performed using MDPI’s Author Services. All changes suggested by the Editor and reviewers were made, as described below:

Reviewer 2

The submission entitled "Transforming properties of the E6/E7 oncogenes from Beta-2 HPV80 in primary human fibroblasts" describes an interesting research work, focusing on the transforming properties from HPV80 in primary fibroblasts. In general, the text is well-written, and the results are sound to the applied methodology and scope of the work.

Major revisions

1.- In Figure 4, the authors mention that Mitomycin was used to suppress cell proliferation; this does not look clear in the figure because in FB-E6/E7- HPV16 model, cell proliferation increase a 30% during 24 h (0h vs 24h). Is necessary to repeat the Wound healing assay with a higher concentration of Mitomycin to stop cell proliferation in all groups.

Response:

Thank you for your comment. Indeed, at the beginning of our experiments, we had to standardize mitomycin concentration. We have previously performed these assays in the E6/E7-HPV16 fibroblasts model as described in PMID: 35276208. For this study, although our interest was only to elucidate the effect of E6/E7 from HPV80, we included the HPV16 model to evaluate both under parallel experimental conditions. We observed that elevated doses lead to fibroblast mortality; thus, the concentration used here was selected to ensure viability while maintaining experimental relevance.

It has been reported that short-term exposure (5 minutes) to Mitomycin C (MMC) at concentrations around 0.4 mg/mL effectively suppresses fibroblast proliferation without compromising cell viability, although higher concentrations could trigger apoptosis (PMID: 23548809), and higher doses induced significant apoptosis in both fibroblasts and HaCaT cells (PMCID: PMC3520454).

Although we agree with you that we use a lower concentration of mitomycin to maintain cell viability, we can observe a difference within fibroblast E6/E7-HPV80 compared to pLVX model. As expected, the fibroblast E6/E7-HPV16 model proliferates much faster.

The quality of these pictures should be improved. It will be interesting if the authors can show the pictures of scratch wound healing assay in higher magnification.

Response:

We thank the reviewer for their observation. We acknowledge that PDF compression reduced the quality of the images in the original submission. In the revised manuscript, we have enhanced the resolution of all figures to ensure optimal clarity. Additionally, we are providing each figure as a separate high-resolution file (TIFF) to further facilitate detailed visualization.

2.- The manuscript is overloaded with references and many of them are unnecessary. It is recommended that the references be removed and updated.

Response:

Thank you for the suggestion. We conducted a thorough review of the discussion and eliminated or replaced some references that we considered not so relevant. We have reduced the number of references from 91 to 86.

3.- Discussion is too long, some reductions are recommended.

Response:

The Discussion section has been revised and shortened as suggested to enhance clarity and focus.

Minor revisions

1.- In introduction, is necessary to explain if E2 protein regulates the expression of the HPV80 E6 and E7 oncogenes during cervical carcinogenesis.

Response:

In our study, we expressed the E6 and E7 genes from HPV80 using a lentiviral vector with a CMV promoter, so native E2 protein regulation was not assessed. Future studies with the complete HPV80 genome will clarify E2's role in E6/E7 regulation for Beta papillomavirus.

2.- In Figure 6 and Figure 7.

If FB-E6/E7-HPV16 model is considered as positive control, ¿Why FB-E6/E7-HPV 80 model presented the most up- or downregulated gene expression in comparison with FB-E6/E7-HPV16?

Response:

Figures 6 and 7 were specifically designed to highlight the genes exhibiting the most significant overexpression or underexpression in the HPV80 model relative to the pLVX control. For comparative purposes, the expression profiles of these same genes in the HPV16 model were also included. HPV16 was selected as a positive control due to its well-characterized status as a high-risk alpha HPV type with established oncogenic potential. Although HPV16 belongs to a different genus than HPV80, it serves as a robust benchmark for evaluating transforming capabilities, given its extensively documented role in cellular transformation.

To enhance the reproducibility and utility of this study, we have included supplementary data containing the transcriptomic dataset, including fold change values and statistical significance for all genes analyzed in both models. Tables in Excel format, Table S1: Differential gene expression analysis of HPV80 (DESeq2 results); Table S2: Differential gene expression analysis of HPV16 (DESeq2 results).

3.- In line 264 (Figure legend), please incorporate “Predicted”.

Response:

As suggested, the word “predicted” has been added to the Figure 9 legend. We also include “predicted” in the Figure 8 legend.

4.- In 4.7 WST-1 Metabolic activity assay (Materials and Methods).

Commonly, Tetrazolium Salt is used in proliferation assays. ¿Why you consider that Tetrazolium Salt is a Metabolic activity assay?

Response:

We appreciate your observation and agree with your comment. WST-1 is a reagent used to quantify indirectly cell proliferation and cellular viability. WST-1 works by measuring the reduction of tetrazolium salts by mitochondrial dehydrogenases into a formazan dye, which relates to the metabolic activity of the cells. Proliferation and viability are often inferred from metabolic activity because more cells or healthier cells will have higher metabolic activity. In our study, we used this term, but we are aware that all these terms can be used.

5.- It is necessary to describe in the 4.12. section if the primers designed for qPCR were analyzed to identify potential secondary structures (self-dimer or hetero-dimer).

Response:

Yes, all primers used for qPCR were analyzed for potential secondary structures, including self-dimers, hetero-dimers, and false priming. All primers used were designed using the Primer-BLAST tool provided by NCBI-NIH (https://www.ncbi.nlm.nih.gov/tools/primer-blast/index.cgi) using as searching Database the Homo sapiens Refseq mRNA. Primer secondary structures were corroborated by the IDT OligoAnalyzer Tool (https://www.idtdna.com/pages/tools/oligoanalyzer). Primer specificity and annealing conditions were determined by conventional PCR. In this new submission, we have included in the 4.12 section this explanation.
